# EEG-Based Schizophrenia Diagnosis through Time Series Image Conversion and Deep Learning

## Dong-Woo Ko and Jung-Jin Yang *

School of Computer Science and Information Engineering, The Catholic University of Korea, 43 Jibong-ro, Bucheon-si 14662, Gyeonggi-do, Korea; koodoowoo@catholic.ac.kr
* Correspondence: jungjin@catholic.ac.kr; Tel.: +82-2-2164-4377

**Abstract:** Schizophrenia, a mental disorder experienced by more than 20 million people worldwide, is emerging as a serious issue in society. Currently, the diagnosis of schizophrenia is based only on mental disorder diagnosis and/or diagnosis by a psychiatrist or mental health professional using DSM-5, a diagnostic and statistical manual of mental disorders. Furthermore, patients in countries with insufficient access to healthcare are difficult to diagnose for schizophrenia and early diagnosis is even more problematic. While various studies are being conducted to solve the challenges of schizophrenia diagnosis, methodology is considered to be limited, and diagnostic accuracy needs to be improved. In this study, a new approach using EEG data and deep learning is proposed to increase objectivity and efficiency of schizophrenia diagnosis. Existing deep learning studies use EEG data to classify schizophrenic patients and healthy subjects by learning EEG in the form of graphs or tables. However, in this study, EEG, a time series data, was converted into an image to improve classification accuracy, and is then studied in deep learning models. This study used EEG data of 81 people, in which the difference in N100 EEG between schizophrenic patients and healthy patients had been analyzed in prior research. EEGs were converted into images using time series image conversion algorithms, Recurrence Plot (RP) and Gramian Angular Field (GAF), and converted EEG images were learned with Convolutional Neural Network (CNN) models built based on VGGNet. When the trained deep learning model was applied to the same data from prior research, it was demonstrated that classification accuracy improved when compared to previous studies. Among the two algorithms used for image conversion, the deep learning model that learned through GAF showed significantly higher classification accuracy. The results of this study suggest that the use of GAF and CNN models based on EEG results can be an effective way to increase objectivity and efficiency in diagnosing various mental disorders, including schizophrenia.

**Keywords:** schizophrenia; EEG; Recurrence Plot; Gramian Angular Field

## 1. Introduction

Schizophrenia is a mental or psychotic disorder characterized by disturbances in cognition, emotional responsiveness and behavior, in which common symptoms include hallucinations, delusions, disorganized speech, abnormal behavior and disturbances of emotion. Schizophrenia is estimated to be experienced by more than 20 million people worldwide, and schizophrenic patients are two to three times more likely to die early than the general population. Statistics show that more than 69% of schizophrenic patients are not receiving appropriate treatment, and more than 90% of these patients live in low-income or middle-income countries with insufficient healthcare access [1,2].

Since chronic schizophrenia patients are difficult to treat, it is imperative that schizophrenia is diagnosed early. Currently, DSM-5, a mental disorder diagnosis and statistical manual published by the American Psychiatric Association, is used by psychiatrists for diagnosis and as a result schizophrenia diagnosis takes an extensive amount of time as it requires long

consultations and observations [3]. Therefore, there is a need for a diagnostic process and system that can objectively and conveniently diagnose schizophrenia.

Many studies have been introduced in order to diagnose various mental disorders, including schizophrenia, and some of these studies use electroencephalography (EEG) [4–7]. EEG is a non-invasive monitoring method, in which electrodes are attached to the scalp surface, measuring the brain's activity status and recording brainwaves. Recordings show generated electrical activity when signals are transmitted between cranial nerves [8]. Since EEG includes neurological conditions that can be leveraged to identify mental illness, recordings are used to distinguish between healthy subjects and patients with mental illness by comparing brain activity. Various studies that focus on the diagnosis and treatment of mental disorders have previously recommended using EEGs, and recently proposed using the field of artificial intelligence in combination with standard diagnostic methods. Most studies that utilize EEGs, use a diagnostic method through artificial intelligence models that use EEGs in the form of graphs or tables, which can limit classification accuracy. In order to ensure high diagnostic accuracy, there is a need for improvement in methodology.

In order to differentiate EEGs of schizophrenic patients from the general public with high accuracy, this study proposed a method of improving diagnostic accuracy by converting EEG, a time series data, into images and then learning in deep learning models. Recent study results showed that a method converting time series data into an image, which was then learned and classified in a deep learning model, demonstrated higher classification accuracy when compared to a method without conversion [9,10]. Based on these insights from prior research, this study used a method of learning deep learning after converting EEGs into images to increase diagnostic accuracy.

## 2. Related Works

There has been a substantial amount of introduced studies that analyze the relationship between EEG and schizophrenia, working towards a more objective and accurate diagnosis of schizophrenia. Ford et al. [11] presented a study to classify schizophrenic patients using EEG through N100, in which brain waves are given auditory stimulation and then stimulation is decreased after 100 ms. Since schizophrenic patients have hearing impairment due to auditory cortex dysfunction, it has been demonstrated that N100, a large, negative evoked potential that is elicited by auditory stimulus, differs in schizophrenic patients compared to the general population. In addition, Kim et al. [12] and Thilakvathi et al. [13] analyzed the pattern of EEG or compared EEG numerical values to validate the correlation between schizophrenia and EEG. Ruxandra et al. [14] trained on a deep learning model without transforming time series data. This study suggests the importance and limitations of learning time series data in deep learning methods.

In Zhang et al. [15], data analyzed in Ford et al. [11] was classified through machine learning technology. The Random Forest was used to differentiate EEGs of schizophrenic patients from the general public, and the highest classification accuracy was 81.1%.

In addition to schizophrenia, the utilization of EEG was also recommended as methods for diagnosing mental disorders such as epilepsy and depression. Archarya et al. [16] proposed a system for automatically diagnosing epilepsy by learning EEG data in the CNN model without any other conversion of EEG, and the accuracy of differentiating between epilepsy patients and healthy subjects was 88.67%.

Alhagry et al. [17] conducted an experiment to discern human emotions using EEG. Emotions were classified by learning raw EEG data in LSTM model, which is used for time series data learning, and the maximum accuracy of emotion classification obtained was 87.99%.

Naira et al. [18] used a new EEG methodology to improve classification accuracy of schizophrenic patients and healthy subjects. This publication presented a method of calculating Pearson Correlation in EEG data, converting into a matrix representing the relationship between EEG channels, and then learning in a CNN model. Previous studies using Support Vector Machine (SVM) and Random Forest obtained 81.07% and 84.5%

accuracy, respectively [19], and deep learning models that learned by converting EEG data into Pearson Correlation matrix demonstrated an improved classification accuracy of 90%.

There are various studies that convert EEG into images to learn from artificial intelligence. WeiKoh et al. [20] used EEG data to limit new methods of diagnosing patients with schizophrenia. In this study, EEG is transformed into a spectrogram and then trained with KNN, one of the machine learning models. Sobahi et al. [21] converted the EEG into an image form using a local binary pattern. Then, using the transformed image, the CNN is trained.

In this research, Recurrence Plot (RP) and Gramian Angular Field (GAF) were used as methods of converting time series data into images [10,22]. The two methods calculated numerical conversion information of time series data using nonlinear analysis and represented the data as a square image. RP and GAF are techniques to which algorithms to analyze patterns of time series data are applied. Therefore, specific changes in EEG data can be efficiently checked, and the overall flow is expressed in one image, which is effective for CNNs where receptive fields are important. In addition, RP and GAF change the patient's EEG into an image for each channel, so it has the advantage of accurately learning a deep learning model with more data.

## 3. Materials and Methods

### 3.1. Data

Data used in this experiment was measured data presented by the National Institute of Mental Health (NIMH; R01MH058262), and was provided at https://www.kaggle.com/broach/button-tone-sz (accessed on 17 July 2022), in which data was measured in 49 schizophrenic patients and 32 healthy patients, generating a total of 81 EEGs. The composition of the research data is as follows.

Data of 81 subjects used in the experiment were measured to study the difference in EEG between schizophrenic patients and healthy subjects. N100 refers to a negative deflection in EEG brain waves after 100 ms when auditory stimulation is given. Schizophrenia patients have problems with N100 because they have hearing impairment due to a dysfunction in the auditory cortex. Thus, a study was conducted to verify the difference in N100 between schizophrenic and healthy patients using EEG data of 81 subjects [12].

EEG was measured 100 times each under the following three conditions.

(1) Subject pressed a button to generate the tone.
(2) Subject passively listened to the same tone.
(3) Subject pressed a button without generating the tone.

EEG was measured in a total of 70 channels by measuring 64-channel scalp EEG and 6 channels around the eyes and nose. In this study that compared the N100 differences between schizophrenic patients and healthy subjects, among the 70 analyzed channels, 9 electrode sites (Fz, FCz, Cz, FC3, C3, CP3, FC4, C4, CP4) had distinct N100 differences. Figure 1 shows the N100 difference between schizophrenic patients and healthy subjects in the Fz channel. As a result of the analysis, in the case of the healthy subjects, N100 was more suppressed in Condition 1 than in Condition 2, but in the case of schizophrenic patients, it was demonstrated that there was no difference between Conditions 1 and 2.

In this experiment, the average of 100 values measured under each condition was used, and the measured data was edited from 1.5 s before hearing the tone to 1.5 s after hearing the tone to use 3 full seconds of EEG data. This experiment proceeded in classifying EEG of schizophrenic patients and healthy subjects using 9 out of 70 channels, similar to that in previous studies.

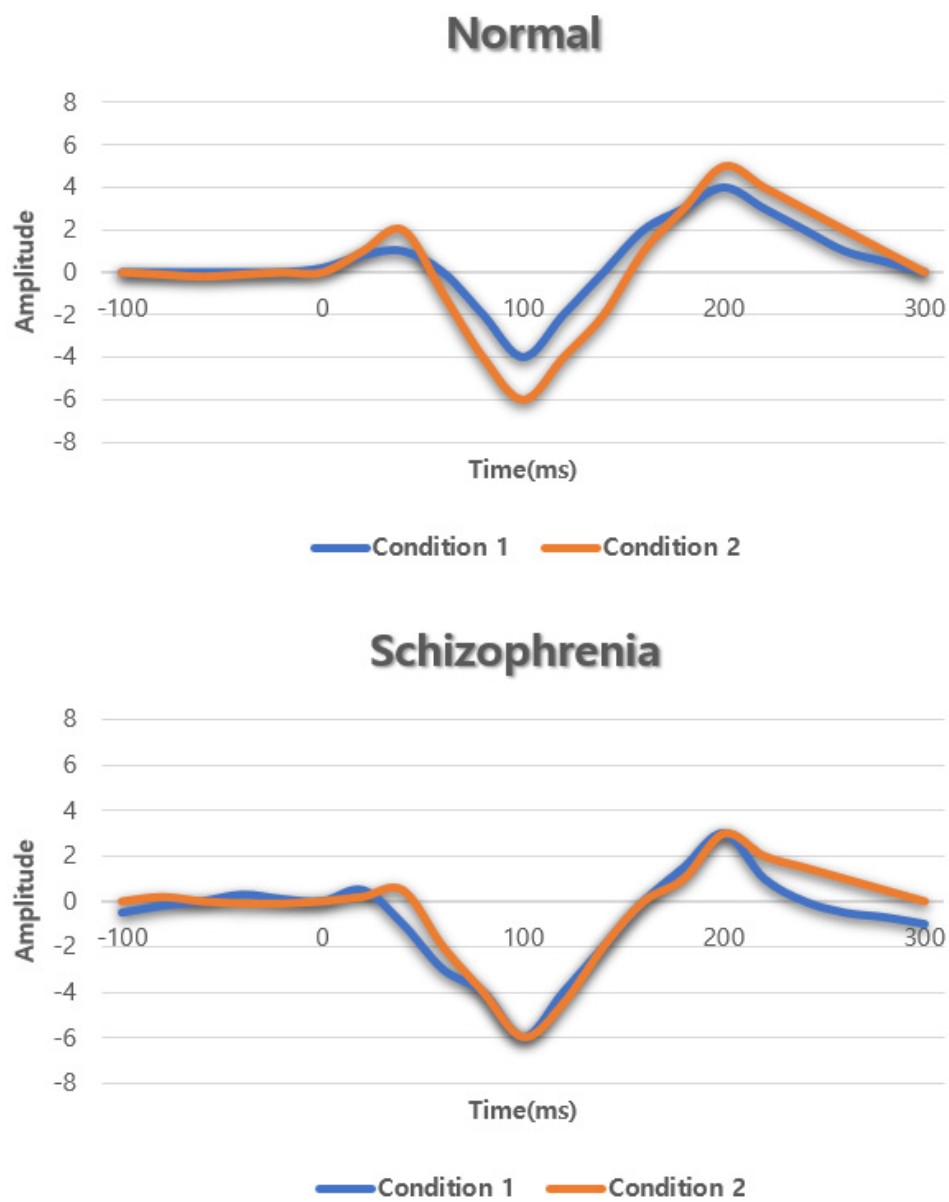

**Figure 1.** Differences in N100 EEG between schizophrenic patients and healthy.

*3.2. Convert Time Series to Images*

Unlike previous studies that learn EEG in table form or in EEG graphs, the newly proposed schizophrenia diagnosis method in this study required a process of converting EEG data into images. Possible methods of converting time series data into images included Recurrence Plot (RP), Gramian Angular Field (GAF), and Markov Transition Field (MTF). The aforementioned methods represent the amount of change in time series data values as a matrix and are expressed as an image. Looking into the characteristics of each method, RP displays the amount of change in the data value in a two-dimensional space, expressing the distance between each point as a matrix, and GAF as the inner matrix of the data values. MTF uses the Markov Transition Matrix to express the probability that the data value will change to the next value in chronological order and converts it into an image.

After comparing classification accuracies of the three image conversion methods, RP and GAF, which have high classification accuracy, were applied to the learning process for this experiment. Figure 2 portrays the overall process of the experiment, and EEGs of schizophrenic patients and of healthy subjects were classified after converting EEG data into images using the previously mentioned two methods and trained in deep learning.

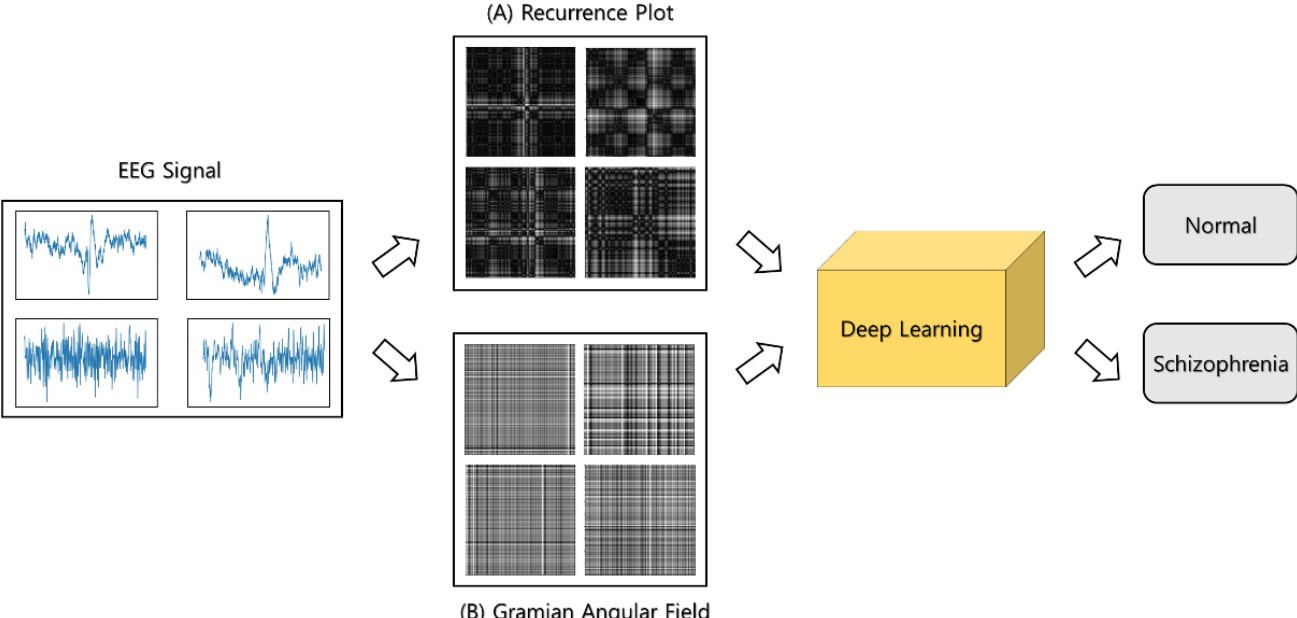

**Figure 2.** Image conversion and deep learning process of EEG data.

### 3.2.1. Recurrence Plot

This step corresponds to (A) of Figure 2 and describes the process of converting the EEG into an image using RP. In order to visualize the movement of time, RPs represent data values in two-dimensional spatial trajectories and then convert them into images by representing the Euclidean distance between spatial trajectories [22]. Therefore, they can be useful in converting time series data, such as EEGs, into images.

$$R_{i,j} := H\left(\varepsilon_i - \|\vec{x}_i - \vec{x}_j\|\right), \ i,j = 1,\dots,N \tag{1}$$

Equation (1) of the Recurrence Plot represented the value of the Euclidean distance matrix. $\|\vec{x}_i - \vec{x}_j\|$ meant the distance between trajectories in a two-dimensional spatial trajectory, and $\varepsilon_i$ meant a threshold. $H(x)$ was the output of a Heaviside Function equal to 1 if $x$ value was greater than or equal to 0 or 0 if $x$ value was less than or equal to 0. If the distance between the trajectories in a two-dimensional spatial trajectory was farther than the threshold, it was represented by a value of 0, and if it was closer than the threshold, it was represented by a value of 1. In the RP, 1 was marked in black and 0 was marked in white to represent an image. Thus, when the EEGs of schizophrenic patients and of healthy subjects were expressed as RPs, the results of the conversion were square images with values of 0 and 1.

### 3.2.2. Gramian Angular Field

The step corresponding to (B) in Figure 2 is the process of converting the EEG into an image using GAF. GAF is an introduced method in order to visualize the movement of time, similar to the RP [10]. If RP is a method of calculating the Euclidean distance between spatial trajectories as a matrix and expressing it as an image, GAF uses polar coordinates to represent the temporal correlation between each time point.

$$\widetilde{x_{-1}} = \frac{(x_i - max(X)) + (x_i - min(X))}{max(X) - min(X)} \tag{2}$$

$$or \qquad \widetilde{x_0} = \frac{x_i - min(X)}{max(X) - min(X)} \tag{3}$$

when *n* time series data values are referred as $X = \{x_1, x_2, \ldots, x_n\}$ values are adjusted to have a value between $[-1, 1]$ or $[0, 1]$ using Equations (2) and (3).

$$\begin{cases} \varnothing = \arccos(\widetilde{x}_i), & -1 \leq \widetilde{x}_i \leq 1, \quad \widetilde{x}_i \in \widetilde{X} \\ r = \frac{t_i}{N}, & t_i \in N \end{cases} \tag{4}$$

Equation (4) $t_i$ refers to time and *N* referred to a constant that normalizes the range of polar coordinates. $\varnothing$ represented the arc cosine value, and *r* the radius value of the polar coordinates. Therefore, the adjusted value $\widetilde{x}_i$ was changed to arc cosine using Equation (4), and time series data was expressed as polar coordinate values by calculating time as the radius. Thereafter, a matrix obtained by calculating the inner product values between each data point was created and displayed as an image.

GAF was also visualized as an image by converting time series data values into matrices, similar to RP, and GAF expressed EEGs of schizophrenic patients and of healthy patients by using inner product values of data to convert into square images. RP and GAF are methods to represent images using changes in time series data. The EEG data used in the experiment is data measuring the difference in EEG change between schizophrenia and normal. Therefore, it can be converted into an image more effectively than the Pearson correlation method expressing the correlation between channels [18], the spectrogram or the local binary pattern method indicating the EEG value [20,21]. Thus, in the next section, we discuss deep learning models for learning RP and GAF.

### 3.3. Deep Learning Using Time Series Images

Since the EEGs of schizophrenic and healthy patients used for schizophrenia diagnosis were nonlinear data, deep learning that distinguishes nonlinear c1haracteristics was used for EEG classification in both groups [23]. Currently, CNN is mainly used in studies that look to distinguish EEG [24,25]. CNN, a high-performing model in image processing, effectively extracts the characteristics of the image, and has a feature that facilitates in model modification [26]. Among the representative models of learning time series data, LSTM, a model that learns by considering the correlation between past and current data, lacks accessibility compared to CNN due to difficult model modification [27]. In this study, the CNN model was used to effectively extract the characteristics of the image by modifying the deep learning model after converting EEG data into an image.

Figure 3A shows the overall structure of the CNN model and Figure 3B shows VGGNet structure used in the experiment. Features of the EEG have significant variation in values, and in order to learn these features in detail, the model was constructed based on VGGNet, which used a small $3 \times 3$ filter [28]. VGGNet is a model that won second place at the 2014 ImageNet Image Recognition Competition, in which 13–16 convolution layers and 3 fully connected layers are used. In the convolution layer of VGGNet, there is an advantage of increasing the processing speed using a $3 \times 3$ filter with a small computational volume, learning data features in detail, and amplifying the effectiveness of nonlinear learning by using the convolution layer several times.

In this experiment, the CNN model used a total of 14 convolution layers, 5 maxpooling layers, 3 fully connected layers, and 1 dropout layer. The input size of VGGNet was $224 \times 224 \times 3$, which was based on an image of an RGB channel with 224 pixels in width and length. However, given that the time series image used in the experiment was a gray scale, only one channel existed. (RP and GAF are images representing patterns of time series images, and are shown in gray scale. A $1 \times 1$ convolution layer using three filters was added on top of VGGNet to make the time series image into three channels so when a one-channel time series image passed through the Convolution layer using three filters, an output value with three channels was obtained.) When a one-channel time series image passed through the convolution layer using three filters, an output value with three channels was obtained, so a convolution layer using three filters was added on top of VGGNet to make the time series image into three channels. After passing through

13 convolution layers and 5 maxpooling layers of VGGNet, the result was inputted to the fully connected layer.

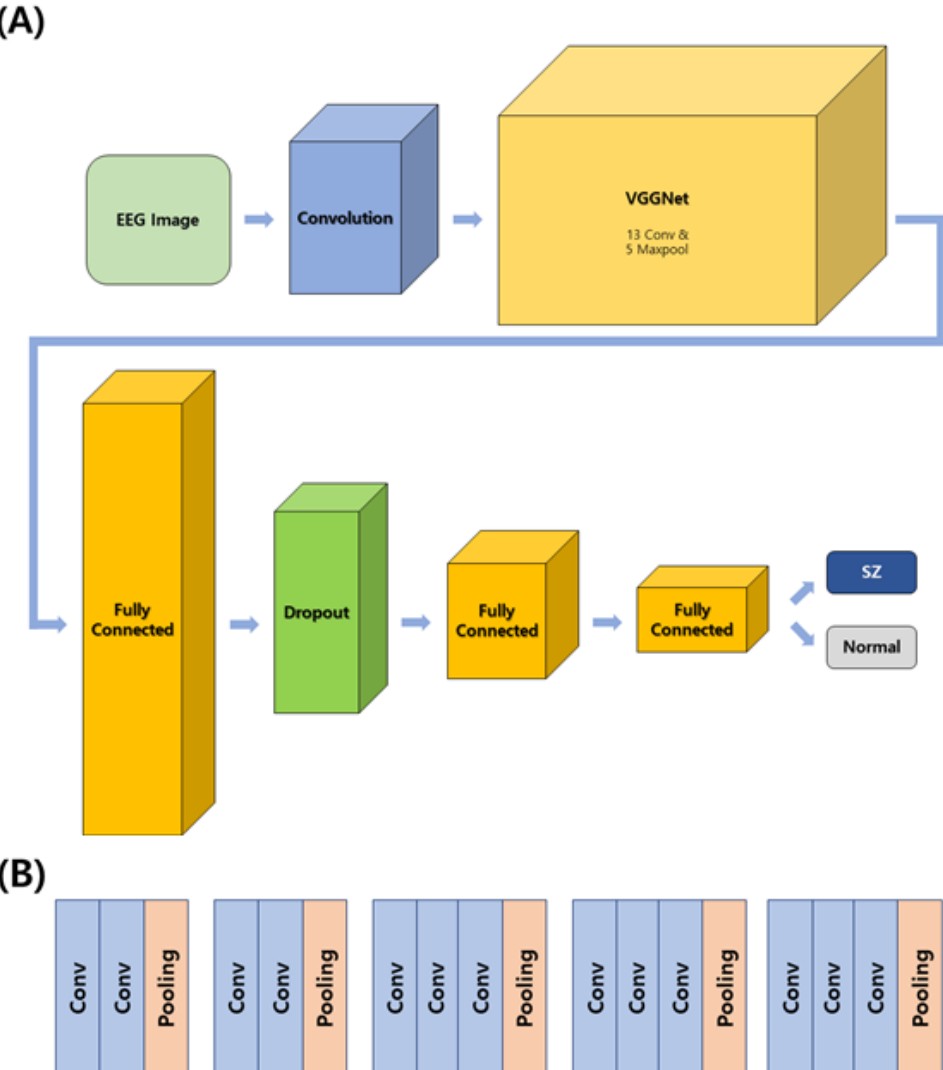

**Figure 3.** VGGNet-based deep learning model (**A**) and VGGNet (**B**) for schizophrenia EEG image learning.

On the 20th to 23rd floors, a fully connected layer that has 4096, 2048, and 1024 units were used. After passing through the first fully connected layer, a dropout layer was used to prevent overfitting. VGGNet used a pre-trained model with ImageNet, and transfer learning was used for this experiment. VGGNet was frozen, convolution and fully connected blocks were trained. Below, the results were compared and analyzed by learning EEG images through the presented model.

## 4. Results

Unlike the existing method of classifying schizophrenic patients by learning deep learning without converting EEG data, this study differentiated schizophrenic patients by learning deep learning after converting time series data EEG into images. A total of 81 EEG data were used in the experiment, in which the data were converted into images using RP and GAF and then learned in the deep learning model.

The measured timeframe of EEG data was 3 s and had 3072 values per channel. Each channel was converted into a single image, and 3072 values passed through RP and GAF to result in a $3072 \times 3072$ size image. Since the input size of the model in this experiment was $224 \times 224$, the $3072 \times 3072$ size image was adjusted to $224 \times 224$ to learn in the deep

learning model. In training, 2187 EEG images of 81 patients were used in 9 channels and 3 conditions were converted to RP and GAF.

RP and GAF are methods for converting time series data into images by calculating the change. Therefore, if RP and GAF are used, the difference in the EEG change between schizophrenia and normal can be expressed as an image. The transformed EEG image is trained on the aforementioned VGGNet-based deep learning model. This model trained the difference between schizophrenia and normal EEG and makes classification possible. Deep learning model trained 200 epochs using binary cross entropy as loss function and Adam as optimizer. Then, the data converted for the experiment is validated by dividing it into train data and test data for each patient using 10-fold cross validation.

Classification accuracy was determined through learning on CNN with EEG data that was converted into images using Recurrence Plot and Gramian Angular Field. Figure 4A,B show the learning curve of the model in which the RP was learned. The maximum accuracy and loss obtained as a result of learning were 0.945, 0.209, respectively. Figure 4C,D show the learning curve of the model in which GAF was trained. The maximum accuracy and loss obtained as a result of learning were 0.963, 0.184, respectively.

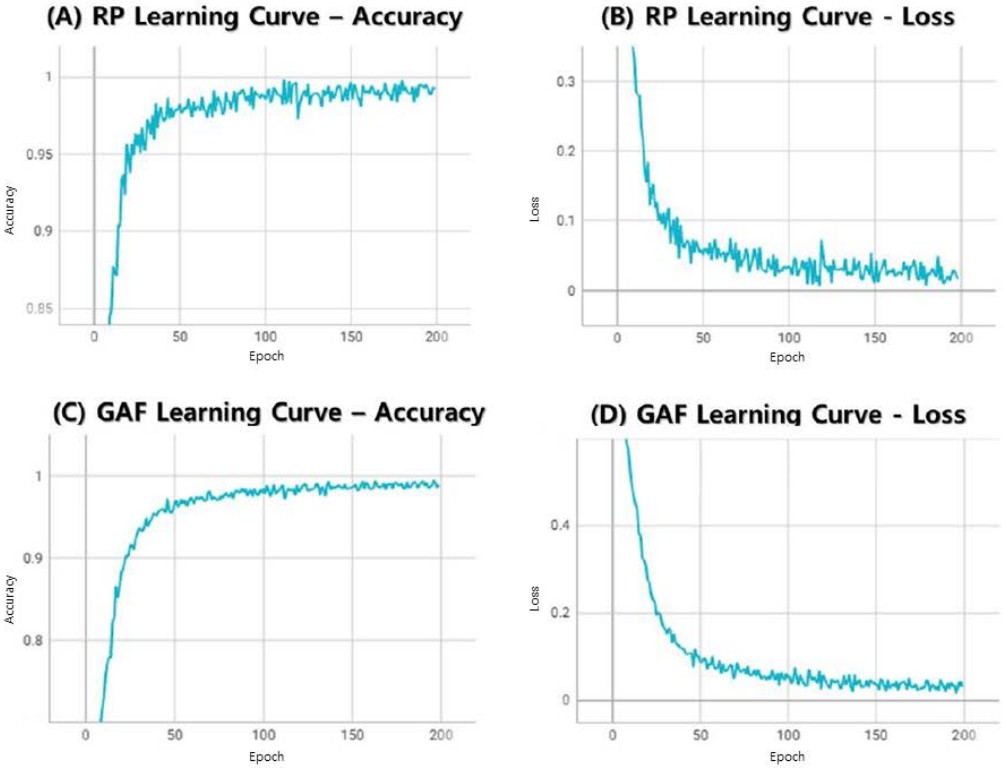

**Figure 4.** Learning curve of Recurrence Plot and Gramian Angular Field.

For result analysis, classification accuracy obtained by learning EEG Graph and classification accuracy from previous studies that differentiated schizophrenic patients using the same data were used in this experiment. Previous studies classified schizophrenic patients using Random Forest but used the same channels as those used in this experiment [16].

Confusion matrices of the approaches using RP and GAF, presented in Tables 1 and 2 were used to determine model performance. For the RP approach, sensitivity and specificity were 90.9% and 88.6%, respectively. For the GAF approach, sensitivity was 93.9% and specificity was 92.1%.

**Table 1.** EEG Recurrence Plot confusion matrix.

|  |  | PREDICTED | |
|---|---|---|---|
|  |  | Schizophrenia | Normal |
| ORIGINAL | Schizophrenia | 1230 | 95 |
|  | Normal | 123 | 739 |

**Table 2.** EEG Gramian Angular Field confusion matrix.

|  |  | PREDICTED | |
|---|---|---|---|
|  |  | Schizophrenia | Normal |
| ORIGINAL | Schizophrenia | 1256 | 67 |
|  | Normal | 82 | 782 |

Table 3 shows the average classification accuracy obtained by learning RP and GAF through the 10-fold cross validation approach. Classification accuracy obtained from learning EEG Graph, from LTSM and from previous studies are presented together for comparison. The EEG graph was trained on the same model as the RP and GAF learning methods to obtain the results, and the LSTM obtained the results by learning the table EEG data without converting it into an image. The classification accuracy obtained by learning EEG Graph was 75.3%, and the classification accuracy from previous studies using same EEG data as our experimental that used Random Forest was 81.1% [15]. In this comparison, the average classification accuracy obtained by learning RP and GAF were 90% and 93.2%, respectively. Thus, improved classification accuracy was achieved compared to prior approaches and previous studies.

**Table 3.** Comparative analysis of learning results of EEG data.

| Method | Accuracy |
|---|---|
| Recurrence Plot | 90.0% |
| Gramian Angular Field | 93.2% |
| EEG Graph | 75.3% |
| LSTM | 66.3% |
| Previous Research | 81.1% |

As a result of data analysis, the method of converting the time series data, EEG, into an image and learning in deep learning indicated an improvement in classification accuracy compared to prior methods and proposed methods in previous studies. In this study, EEG data was converted into images using RP and GAF, and based on the analysis results, it can be demonstrated that the approach using GAF is more effective and accurate in classifying EEGs in schizophrenic patients and healthy subjects.

## 5. Discussion

An increasing amount of proactive research for mental disorder diagnosis is being conducted as the number of patients suffering from mental disorders has had a rapid upward trend. Currently mental disorder diagnosis depends only on a diagnosis by a psychiatrist using DSM-5, a statistical manual. Societies with poor access to healthcare, in particular, face challenges in inaccurate diagnosis in addition to difficulties in early diagnosis. It is necessary to improve the accuracy by supplementing the limited approach of existing research. In this study, a new methodology was proposed to improve the efficiency and accuracy of mental disorder diagnosis.

Unlike conventional methods of learning EEGs of schizophrenic and healthy patients in deep learning in the form of graphs or tables, this study used a different approach of converting EEGs into images and diagnosing schizophrenic patients in deep learning models. Recurrence Plot and Gramian Angular Field were used to convert time series data into images, while deep learning models used CNN models built on VGGNet.

EEG data of 81 subjects were converted into images, then learned in a deep learning model. Results from this experiment were analyzed and compared to previous studies that used a variety of approaches including learning EEG graphs in deep learning, learning in LSTM, which is used for time series data learning, and the NIMH experiment in which the EEG data used in this experiment was originally collected.

Demonstrated classification accuracy was 66.3% from learning using LSTM, 75.3% in the experiment that used learning EEG graphs of 81 subjects, and 81.1% in a previous study that classified EEG using Random Forest. The classification accuracy obtained by learning RP and GAF was 90.0% and 93.2%, respectively, which was a 12.1% improvement from the classification accuracy of existing methods and previous studies. Additionally, a comparative analysis of the learning results from RP and GAF confirmed that the use of GAF as a method for diagnosing schizophrenia was more effective than the use of RP.

This study proposed an approach for effectively diagnosing schizophrenia. The current diagnosis method of schizophrenia on mental disorder diagnosis by a psychiatrist using DSM-5, a statistical manual. Furthermore, there are challenges in early diagnosis given that patients must have extensive in-person visits at mental health institutions for medical treatment. As such, the method presented in this study provides a potential procedure for an objective and effective diagnosis of schizophrenia. Additionally, leveraging this proposed method further supports the possibility of diagnosing mental disorders for patients in socio-economically disadvantaged societies where healthcare access is difficult, for patients with frequent outpatient treatment and can be expanded to develop convenient patient management for psychiatrists.

Future research will focus on how to further improve the accuracy of schizophrenia classification by supplementing the deep learning model used in this experiment. It can be seen that this study and improved methodologies have the potential to be expanded and applied to various mental disorders other than schizophrenia, such as depression, impulse control disorder, and anxiety disorder, in which diagnosis is solely dependent on the utilization of DSM-5.

**Author Contributions:** Conceptualization, D.-W.K. and J.-J.Y.; methodology, D.-W.K. and J.-J.Y.; software, D.-W.K.; validation D.-W.K. and J.-J.Y.; formal analysis, D.-W.K. and J.-J.Y.; investigation, D.-W.K.; resources, D.-W.K. and J.-J.Y.; data curation, D.-W.K.; writing—original draft preparation, D.-W.K.; writing—review and editing, D.-W.K. and J.-J.Y.; visualization, D.-W.K.; supervision, J.-J.Y.; project administration, D.-W.K.; funding acquisition, J.-J.Y. All authors have read and agreed to the published version of the manuscript.

**Funding:** This work was supported by a Global Research Laboratory (GRL) Program of the National Research Foundation of Korea (NRF) grant funded by the Korean government (Ministry of Science and ICT) (NRF-2016K1A1A2912755). This work was supported by the Catholic University of Korea, Research Fund, 2019.

**Institutional Review Board Statement:** Data were initially collected with funding from the National Institute of Mental Health (NIMH grant number R01MH058262).

**Informed Consent Statement:** All study participants gave written, informed consent to participate in this study, which received Institutional Review Board approval (NIMH project number: R01MH058262).

**Data Availability Statement:** https://www.kaggle.com/broach/button-tone-sz (accessed on 17 July 2022).

**Conflicts of Interest:** The authors declare no conflict of interest.

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
