# Peer review of "EEG-Based Schizophrenia Diagnosis through Time Series Image Conversion and Deep Learning"

_electronics, doi:10.3390/electronics11142265_

Round 1

Reviewer 1 Report

The article proposes the application of a CNN based on VGGNet for the classification of EEG time series data transformed into images via Recurrence plot and Gramian Angular Field. The presentation of the methodology generally flows well. Related work should be a separate section from Introduction. The related work should also present recent applications of deep learning models to time series directly, without transformation to images (e.g. see https://doi.org/10.1145/3531326 , https://doi.org/10.1371/journal.pone.0236401) and provide a rationale why it is better to make the transformation into images.

In it not clearly stated how many images are produced from the EEG data of the 81 patients. Also, the separation between training and validation should be made at the level of the patients, as well, not only at the level of images, meaning that images from the same patient should not appear both in training and validation sets.

Figure 4 needs axes labels for all the plots.

The caption from Table 3 is the one from the template. Please explain how the results in Table 3 are obtained. Did the authors implement the other techniques and the same experimental setup is used? What is “Previous Research”?

Line 98, please erase “The machine learning technology” and start the phrase with “Random forest”.

Please put a space between citations and the words near them.

Reviewer 2 Report

This study presents a method for the detection of schizophrenia from EEEG signals by means of the transformation of EEG time series into RP and GAF images and their classification in a VGGNet-based CNN model. I would ask authors to check the following comments in order to improve the content of the manuscript:

1.- The conversion of EEG signals into images for schizophrenia detection with CNN classification models is an interesting approach. However, there are some recent works that have also dealt with this problem. Please, a thorough review of the state of the art, especially about the detection of schizophrenia with EEG signals and deep learning algorithms in the last years should be done. Here are some examples:

- Bagherzadeh, S., Shahabi, M. S., & Shalbaf, A. (2022). Detection of schizophrenia using hybrid of deep learning and brain effective connectivity image from electroencephalogram signal. Computers in Biology and Medicine, 105570.

- Korda, A. I., Ventouras, E., Asvestas, P., Toumaian, M., Matsopoulos, G. K., & Smyrnis, N. (2022). Convolutional neural network propagation on electroencephalographic scalograms for detection of schizophrenia. Clinical Neurophysiology, 139, 90-105.

- WeiKoh, J. E., Rajinikanth, V., Vicnesh, J., Pham, T. H., Oh, S. L., Yeong, C. H., ... & Cheong, K. H. (2022). Application of local configuration pattern for automated detection of schizophrenia with electroencephalogram signals. Expert Systems, e12957.

- Sobahi, N., Ari, B., Cakar, H., Alcin, O. F., & Sengur, A. (2022). A New Signal to Image Mapping Procedure and Convolutional Neural Networks for Efficient Schizophrenia Detection in EEG Recordings. IEEE Sensors Journal22(8), 7913-7919.

2.- In line with the previous question, authors should highlight the novelty of this work with respect to similar recent studies.

3.- In this method, each EEG channel is converted into an image. Are all channels used simultaneously for the learning of the CNN algorithm? Or are the evaluated separately? It could be possible that different channels may reveal different information. If all channels are analyzed at the same time, these differences cannot be detected.

4.- VGGNet is a pretrained CNN. How authors have adapted this pretrained model to their problem? The fine-tunning process should be explained.

5.- It is not clear for me why three filters are used on top of VGGNet. This process should be better explained.

6.- It is said that only nine EEG channels from 81 subjects were analyzed. Each channel was converted into an image, hence 9*81=729 images. However, the total number of samples in the confusion matrices is 2187 (729*3). Please, indicate the total number of images that were introduced in the CNN model and how they were obtained. Also, justify if this number of images is enough for a proper implementation of a CNN model, for which huge datasets with thousands of images are needed.

7.- Only training and test results are presented, and it is not said if any method like cross-validation was used for overfitting prevention. It seems that the model was trained and tested with the same samples. In addition, the results should be validated with another dataset different from the samples used for training and testing. The training-test-validation process should be thoroughly revised.

8.- In my opinion, the structure of the contents should be revised:

a) In Introduction section, I would say that lines 70-86 could be better in Discussion or Conclusions section, as this information is not an introduction to the background of the problem.

b) Also in Introduction section, maybe lines 87-117 could be included in a new subsection “State of the art”.

c) In Results section, lines 281-326 are a comparison of the results obtained with previous studies and discussion of the novelties/conclusions of this work. Therefore, this information should be included in Discussion or Conclusions section instead of in Results section.

d) Part of the information in Discussion section is redundant with the aforementioned lines. Please correct.

e) Some information is redundant along the manuscript. Please revise and eliminate redundant information.

9.- In Table 3, please include the references of the works mentioned for comparison with the results obtained with the proposed method.

10.- Please change the caption of Table 3 (line 295) and correct some typos along the text.

Reviewer 3 Report

The novelty of the proposed work needs to be highlighted providing more details in methods and material, and results. 

Reviewer 4 Report

Dear Authors,

The proposed work is good and it needs a few improvements.

I request you to refer to the following related paper which considered a similar methodology to examine the EEG signal:  Implementation of deep neural networks to classify EEG signals using gramian angular summation field for epilepsy diagnosis

I request you to consider the following suggestions:

1. In the introduction section, please discuss the limitations in  signal analysis and the need for the Signal-Image conversion. At the end of the introduction section, list the contribution of the proposed work.

2. Include a section "Related Works" and discuss the similar works available in the literature (Ex: Various Signal-Image conversion methods). and the merit of Gramian Angular Field (GAF) compared to other methods.

3.  In Fig 2, two plots are depicted. Please discuss the merits of GAF compared to the recurrence plot. Further, these images are obtainable in RGB form, why the gray scale is considered? Please discuss.

3. Figure 3 can be enhanced by including the actual VGGNet compared to a simple block diagram. I request you to improve it.

4. In section 3, the results showed (Accuracy and Loss ) can be improved by considering other results, such as the Confusion matrix and ROC curve.

5.  In Table 3, the considered approach provided an accuracy of 93.2%. Hope it is obtained by considering the gray-scale version of the GAF. If the RGB scale image is considered, it can be less. Please confirm. Also, along with the accuracy, I request to include other methods like;  Precision, Sensitivity, and Specificity.

6. I request you to consider the following Signal-Image conversion methods and discuss their merits and demerits in "Related Works": 

a. Detecting epilepsy in EEG signals using synchro-extracting-transform (SET) supported classification technique

b. Automated detection of schizophrenia using nonlinear signal processing methods

c. Application of local configuration pattern for automated detection of schizophrenia with electroencephalogram signals

Round 2

Reviewer 2 Report

After the changes included by the authors, in my opinion there are still many weak points that have not been solved. Although some modifications have been made, the solutions proposed by the authors to my comments are not enough for completely correcting the manuscript as the way I think it should have been improved. The explanations given by the authors do not provide extra information about the methods and the procedures, especially regarding the images processing and classification step. In addition, I still cannot see the novelty of this study with respect to previous works in the literature, which I think is the most serious negative point in this review.

Round 3

Reviewer 2 Report

Authors have met most of my requirements. In my opinion, the manuscript can be accepted for publication.

This manuscript is a resubmission of an earlier submission. The following is a list of the peer review reports and author responses from that submission.